# ODE-Constrained Generative Modeling of Cardiac Dynamics for 12-Lead ECG Synthesis

**Yakir Yehuda**                                                        *y.yakir@cs.technion.ac.il*
*Department of Computer Science*
*Technion - Israel Institute of Technology*

**Kira Radinsky**                                                       *kirar@cs.technion.ac.il*
*Department of Computer Science*
*Technion - Israel Institute of Technology*

Reviewed on OpenReview: *https://openreview.net/forum?id=4N56Pwwsti*

## Abstract

Generating realistic training data for supervised learning remains a significant challenge in artificial intelligence, particularly in domains where large, expert-labeled datasets are scarce or costly to obtain. This is especially true for electrocardiograms (ECGs), where privacy constraints, class imbalance, and the need for physician annotation limit the availability of labeled 12-lead recordings, motivating the development of high-fidelity synthetic ECG data. The primary challenge in this task lies in accurately modeling the intricate biological and physiological interactions among different ECG leads. Although mathematical process models have shed light on these dynamics, effectively incorporating this understanding into generative models is not straightforward. We introduce an innovative method that employs ordinary differential equations (ODEs) to enhance the fidelity of 12-lead ECG data generation. This approach integrates cardiac dynamics directly into the generative optimization process via a novel Euler Loss, producing biologically plausible data that respects real-world variability and inter-lead constraints. Empirical analysis on the G12EC and PTB-XL datasets demonstrates that augmenting training data with MultiODE-GAN yields consistent, statistically significant improvements in specificity across multiple cardiac abnormalities. This highlights the value of enforcing physiological coherence in synthetic medical data.

## 1 Introduction

The generation of synthetic 12-lead electrocardiogram (ECG) data has emerged as a significant area of research, driven by the need for large and diverse datasets to train machine learning models for various medical applications (de Melo et al., 2022). ECG data, which records the electrical activity of the heart, is critical for diagnosing and monitoring cardiac conditions. While real ECG datasets are available, they are often biased toward common conditions, exhibit severe class imbalance for rare disorders, and are constrained by privacy and data security concerns (Voigt & Bussche, 2017). Moreover, acquiring annotated multi-lead ECG data at scale requires physician review, making large-scale expert labeling costly and further limiting the availability of rare cardiac disorder cases. Generating synthetic data offers a promising solution to address privacy concerns associated with the distribution of sensitive health information (Giuffrè & Shung, 2023). In this context, ensuring that synthetic signals are physiologically realistic, not just visually plausible, is essential for their utility in clinical and machine learning contexts, especially for tasks such as anomaly detection and disease classification. Recent advances suggest that embedding domain-specific mathematical models into deep generative models can lead to more faithful data representations. In this work, we leverage such mathematical models by incorporating ordinary differential equations (ODEs) that describe cardiac dynamics directly into the generative process.

ODE-based models are well-established in the sciences for simulating complex systems, enabling nuanced manipulation and interpretation of variables over time. Realistic heart models, for example, require extensive knowledge of cardiac mechanics and anatomy to simulate phenomena such as pressure–volume relationships and various pathologies. In contrast, our work focuses on the ECG Dynamical Model (EDM), which captures electrophysiological dynamics via a system of ODEs McSharry et al. (2003). Embedding this model within our generative framework enables continuous simulation and synthesis of ECG signals with enhanced physiological fidelity.

To this end, we introduce MultiODE-GAN, a novel Generative Adversarial Network (GAN) framework specifically designed for generating synthetic 12-lead ECG data. This framework is meticulously tailored to replicate cardiac cycles from ECG signals across all 12 leads. Our GAN not only employs an adversarial loss to encourage realistic outputs, but also introduces an *Euler Loss* that quantifies the discrepancy between the temporal dynamics of generated heartbeats and those predicted by the EDM. This loss enforces two levels of physiological consistency: (1) individual leads follow the ODE-prescribed dynamics, and (2) inter-lead dependencies (e.g., Einthoven's law), which is essential for synthesizing coherent 12-lead ECG signals and capturing both temporal and spatial dependencies. Since the ODEs cannot generally be solved analytically, we leverage the Euler method to numerically evaluate this loss. We validate MultiODE-GAN on gold-standard 12-lead ECG datasets and show that incorporating ODE-based constraints into the generative process significantly enhances the physiological realism of synthetic signals. To assess clinical utility, we further evaluate MultiODE-GAN in collaboration with the Interventional Cardiology Unit for early detection of left ventricular systolic dysfunction (LVSD) using real patient data. Additionally, classifiers trained on synthetic ECGs generated by MultiODE-GAN show improved performance in downstream heartbeat classification tasks, demonstrating its value for anomaly detection and early screening applications. In a retrospective clinical study (Appendix B), the method achieves specificity improvements comparable to those of board-certified cardiologists for early LVSD detection, highlighting its potential to support clinical decision-making.

Our contributions are threefold: (1) We propose MultiODE-GAN, a novel ODE-constrained GAN that incorporates a physiological heart model and explicit inter-lead dependencies for synthetic 12-lead ECG generation. (2) We provide extensive empirical evidence, showing that MultiODE-GAN consistently improves downstream 12-lead ECG classification performance across multiple abnormalities and benchmarks. (3) We release our implementation as open source to support reproducibility and further research in medical AI.

## 2  Related Work

The availability of annotated data often represents a major bottleneck in the development of deep learning models (de Melo et al., 2022). Unlike biological systems, deep learning methods still lack capabilities for richer representations and sophisticated learning, which synthetic data and simulators can help to address (Cichy & Kaiser, 2019). Synthetic data plays an essential role in training deep neural networks across domains such as detection, segmentation, and classification (Tremblay et al., 2018; Ros et al., 2016; de Melo et al., 2022).

In the medical and machine learning communities, the generation of synthetic ECG data has gained significant interest due to the critical need for large, high-quality datasets. Various generative models have been proposed, each with unique approaches to simulate realistic ECG signals.

A cornerstone of recent advancements is the use of physiological mathematical models, such as partial differential equations, to represent biomedical phenomena including electrical activation in the heart (Boulakia et al., 2010) and neuronal activity and disorders (Fenwick et al., 1971; Hallez et al., 2007). A notable advancement in ECG generation is the dynamical models based on ordinary differential equations (ODEs). McSharry et al. (2003) proposed a set of coupled ODEs to simulate the heart's electrophysiological activity providing a physiologically grounded approach to ECG generation.

The advent of deep learning has introduced innovative methods for generating synthetic ECG data, among which Generative Adversarial Networks (GANs) (Goodfellow et al., 2014) stand out. GANs, through adversarial training of a generator and discriminator, have been effectively employed to create high-fidelity

synthetic data (Frid-Adar et al., 2018; Karras et al., 2017). Variants such as DCGAN (Radford et al., 2016). In our research, we utilize WaveGAN (Donahue et al., 2018), a GAN variant designed for waveform data, making it particularly well-suited for modeling the temporal dynamics of ECG signals.

Research by Golany et al. (2020) demonstrated that GANs can generate realistic single-lead ECG heartbeats using an ODE-based simulator and suggested the potential for extending this idea to multi-lead ECG generation. However, SimGAN is inherently limited to single-lead dynamics and cannot capture the essential physiological interdependencies between the 12 standard ECG leads. We extend this line of work by introducing physiologically motivated constraints that enforce anatomical relationships (e.g., Einthoven's law) across all 12 leads. Unlike purely data-driven approaches such as SSSD-ECG (Alcaraz & Strodthoff, 2023), MultiODE-GAN explicitly embeds these biophysical laws into the training loop. Applying SimGAN independently to each lead is insufficient, as it ignores these critical inter-lead dependencies. In Section 6, we empirically demonstrate the shortcomings of such single-lead inference approaches.

Previous studies have explored various methods for generating 12-lead ECG data. Liu et al. (2020) suggested the use of vector quantized variational autoencoders (VQ-VAE) for generating new samples to enhance the training of a 12-lead ECG classifier. Kong et al. (2021) introduced DiffWave, a versatile diffusion probabilistic model designed for both conditional and unconditional waveform generation.

Huang et al. (2023) proposed an unsupervised GAN-based method for generating noisy ECG signals without requiring labeled data. More recently, (Alcaraz & Strodthoff, 2023) introduced SSSD-ECG, a diffusion-based model for 12-lead ECG generation that represents the current state-of-the-art. We compare our approach, MultiODE-GAN, against SSSD-ECG and show that MultiODE-GAN achieves state-of-the-art performance in generating physiologically realistic 12-lead ECGs.

To demonstrate the impact of synthetic data, we also include a 12-lead ECG classifier trained on both real and synthetic data. Research has shown that deep learning has revolutionized ECG classification, achieving state-of-the-art results (Ribeiro et al., 2020; Attia et al., 2019; Nejedly et al., 2021) and, in some cases, outperforming cardiologists (Golany et al., 2022; Choi et al., 2022). However, these models often require extensive labeled datasets and do not inherently incorporate underlying cardiac physiology, a gap our work aims to fill by bridging dynamic physiological models with advanced machine learning techniques.

Our work builds on prior efforts to bridge dynamic physiological modeling and generative approaches for 12-lead ECG synthesis, particularly in capturing inter-lead dependencies. We advance this direction by directly embedding an ODE-based model into the generative process, guided jointly by physiological principles and data-driven learning. We adopt a GAN backbone for its computational efficiency on high-frequency waveforms and its compatibility with differentiable ODE-based penalties, extending these constraints to diffusion models is an exciting direction for future work.

## 3 Physiology-Based ECG Dynamical Model (EDM)

The ECG Dynamical Model (EDM), originally introduced by McSharry et al. (2003), employs a system of three coupled ordinary differential equations (ODEs) to model the prototypical morphology of the ECG waveform. The model generates a trajectory in three-dimensional space $(x(t), y(t), z(t))$, where the path is modulated at key points corresponding to the $P$, $Q$, $R$, $S$, and $T$ waves. Each of the three components, the $P$ wave, $QRS$ complex, and $T$ wave contributes to the prototypical electrical activity of the heart. Notably, the EDM can simulate synthetic ECGs with physiologically plausible $PQRST$ morphology and prescribed heart rate dynamics, parameterized by mean/standard deviation and the spectral characteristics of heart rate variability.

### 3.1 Model Formulation

The EDM breaks the ECG down into three major components: the $P$ wave, $QRS$ complex, and $T$ wave. These correspond to distinct phases of the cardiac cycle, creating the canonical $PQRST$ sequence. The model is defined by three coupled ODEs that evolve the trajectory $(x(t), y(t), z(t))$:

$$\dot{x} = \alpha x - \omega y \equiv f_x(x, y; \eta) \tag{1}$$

$$\dot{y} = \alpha y + \omega x \equiv f_y(x, y; \eta) \tag{2}$$

$$\dot{z} = -\sum_{i \in \{P,Q,R,S,T\}} a_i \Delta\theta_i \cdot \exp\left(-\frac{\Delta\theta_i^2}{2b_i^2}\right) - (z - z_0(t))$$

$$\equiv f_z(x, y, z, t; \eta) \tag{3}$$

where:

- $a_i$, $b_i$, $\theta_i$: amplitude, width, and angular centers of the Gaussian-shaped $P$, $Q$, $R$, $S$, $T$ waves.

- $\omega = 2\pi f$ is the angular velocity, where $f$ represents the heart rate. This parameter links the trajectory's rotation around the limit cycle to the beat-to-beat heart rate.

- $\alpha$ is defined as $\alpha = 1 - \sqrt{x^2 + y^2}$. This term drives the trajectory towards an attracting limit cycle.

- $\Delta\theta_i = (\theta - \theta_i) \bmod 2\pi$, where $\theta = \arctan 2(y, x) \in [-\pi, \pi]$ is the phase angle of the trajectory. As the $z$ trajectory approaches one of the fixed angular positions $\theta_i$.

- $z_0(t) = A \sin(2\pi f_2 t)$: baseline wander driven by respiratory frequency $f_2$.

We collect the ODE parameters as $\eta = \{\theta_i, a_i, b_i \mid i \in \{P, Q, R, S, T\}\}$.

Each of the 12 leads, along with each type of abnormality, is parameterized by a unique set of EDM coefficients, available in our public code repository. The solution to the EDM system describes a continuous trajectory in the three-dimensional state space $(x, y, z)$, where the $z(t)$ component directly corresponds to the synthesized ECG signal.

We estimate the EDM parameters $\eta_k^c$ for each lead $k$ and abnormality class $c$ by fitting the McSharry model (McSharry et al., 2003) to real heartbeats using a least-squares optimization procedure. For each (lead, class) pair, we fit the parameters on training beats only and keep $\eta_k^c$ fixed during GAN training.

The characteristic $P$, $Q$, $R$, $S$, and $T$ waves are determined by the parameters $\theta_i$, which fix their positions on the unit circle. As the trajectory approaches each $\theta_i$, the $z$ component is transiently elevated or depressed, corresponding to the onset of a cardiac wave. The amplitude and duration of each waveform are precisely controlled by parameters $a_i$ and $b_i$, respectively, enabling fine-grained manipulation of ECG morphology to reflect a wide range of physiological and pathological conditions.

### 3.2 Numerical Integration of the EDM

The ordinary differential equations (ODEs) that form the basis of our ECG Dynamical Model are solved numerically using methods from the Runge–Kutta family (Butcher & Butcher, 1987). For this purpose, we employ the Euler discretization as a differentiable local consistency operator for enforcing ODE compliance, with a fixed time step $\Delta t = \frac{1}{f_s}$, where $f_s$ is the sampling frequency rather than as a high-accuracy numerical solver (Atkinson, 2008) . The time step $\Delta t$ (1/500 Hz) is sufficiently small to avoid numerical instability or stiffness issues, in practice, higher-order solvers (e.g., Runge–Kutta 4) produce trajectories that are nearly identical for this system.

The Euler method employs a finite difference approximation, as detailed by (Milne-Thomson, 2000):

$$\frac{du}{dt}(t) \approx \frac{u(t + \Delta t) - u(t)}{\Delta t}$$

For a generic ODE expressed as $\dot{u} = F(t, u)$, the Euler method approximates the solution iteratively as:

$$u(t + \Delta t) \approx u(t) + F(t, u(t))\Delta t \tag{4}$$

We implement this method by applying Eq. (4) iteratively across $L$ time-steps, where each time-step $\ell$ corresponds to $t_\ell = \ell \Delta t$, with $\Delta t = \frac{1}{f_s}$ fixed by the sampling frequency $f_s$.

In the discrete formulation, for a time-step $\ell$ corresponding to $t_\ell = \ell \Delta t$, the iterative update is given by:

$$u_{\ell+1} = u_\ell + f(\ell, u_\ell)\Delta t \tag{5}$$

Applying this scheme to the EDM equations (Equations 1, 2, and 3), we derive the discrete trajectories $(x, y, z)$ across successive time-steps:

$$
\begin{aligned}
t_\ell &= \ell \Delta t \\
x_{\ell+1} &= x_\ell + f_x(x_\ell, y_\ell; \eta)\Delta t \\
y_{\ell+1} &= y_\ell + f_y(x_\ell, y_\ell; \eta)\Delta t \\
z_{\ell+1} &= z_\ell + f_z(x_\ell, y_\ell, z_\ell, t_\ell; \eta)\Delta t
\end{aligned}
\tag{6}
$$

This approach ensures that each step forward in time $\ell + 1$ is calculated based on the known values from the previous step $\ell$, requiring the initial conditions to start the simulation.

## 4 MultiODE-GAN

We propose MultiODE-GAN, a generative adversarial framework that incorporates physiological constraints from the ECG Dynamical Model (EDM), as described in Section 3. While traditional GANs often struggle to capture the intricate morphology and physiological consistency required for realistic medical signals, MultiODE-GAN addresses this by embedding ODE-based dynamics directly into the generation process.

### 4.1 MultiODE-GAN Framework

We implement an extended SimGAN baseline by applying the original single-lead ODE-based loss independently to each of the 12 leads, without modeling inter-lead dependencies. This provides a controlled comparison to our method. SimGAN (Golany et al., 2020) was originally designed to generate only a single ECG lead and, by construction, cannot model the physiological coupling that governs the full 12-lead system. In contrast, MultiODE-GAN introduces a novel multi-lead ODE-constrained formulation that embeds both intra-lead dynamics and physiologically grounded inter-lead relationships directly into the generator's loss. This enables the model to capture the spatial-temporal structure of the 12-lead ECG by explicitly incorporating ODE-based constraints alongside Einthoven's law, producing substantially more realistic and clinically coherent synthetic signals.

For the underlying architectural backbone, we adopt WaveGAN (Donahue et al., 2018), a one-dimensional adaptation of DCGAN (Radford et al., 2016) well-suited for waveform generation. Although originally designed for single-channel output, we extend WaveGAN to natively produce multi-channel signals, enabling full 12-lead ECG synthesis. We further augment the architecture with an EDM-based Euler residual loss that injects biophysical structure into the generator. The key innovation of MultiODE-GAN lies in embedding these ODE-based physiological constraints directly within the GAN training loop, allowing the model to enforce both the dynamics of individual leads and the physiological relationships between them.

We segment each ECG recording into individual cardiac cycles (heartbeats), represented as fixed-length matrices $h \in \mathbb{R}^{12 \times L}$, where $L$ denotes the duration of a single beat as defined by the RR interval. Details of the segmentation process are described in Section 5.

We adopt the Wasserstein GAN with gradient penalty (WGAN-GP) (Gulrajani et al., 2017) as the core adversarial loss formulation, chosen for its improved training stability and ability to avoid mode collapse compared to vanilla GANs, particularly beneficial in high-dimensional waveform domains.

The Wasserstein loss is defined as:

$$L_{WGAN}(D_w, G) = \mathbb{E}_{h \sim p_{data}}[D_w(h)] - \mathbb{E}_{z \sim p_z(z)}[D_w(G(z))] \tag{7}$$

Here, $p_{data}$ denotes the distribution of real ECG heartbeats, and $p_z(z)$ represents the prior distribution of noise inputs. The discriminator $D_w$ is trained not to classify inputs as real or fake, but to estimate the Wasserstein distance between real and generated distributions. This formulation helps the generator $G$ converge more reliably toward producing physiologically realistic ECG signals.

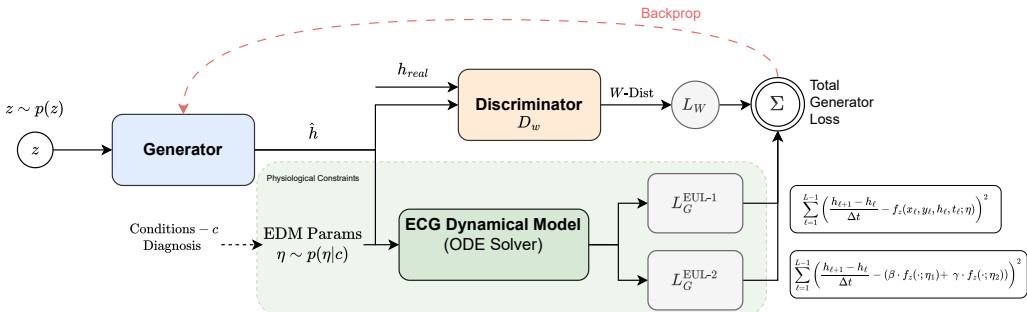

Figure 1: The MultiODE-GAN generator architecture. It takes random noise input to produce synthetic 12-lead ECG heartbeats. The generator's total loss combines the adversarial Wasserstein loss with two novel physiological constraints from the ECG Dynamical Model (EDM): (1) an intra-lead Euler loss enforcing alignment with the ODE-based biophysical model, and (2) an inter-lead Euler loss ensuring consistency across related leads.

## 4.2 MultiODE-GAN Generator

The MultiODE-GAN Generator enhances the WaveGAN architecture (Donahue et al., 2018) by integrating dynamics from the ECG model described in Section 3. While retaining the architectural foundation of WaveGAN, we extend the loss function to better capture the physiological accuracy required for realistic ECG signal generation.

**Generator Loss Function:** The generator's loss combines the Wasserstein distance with an EDM-derived Euler Loss that enforces physiological consistency (Section 3). This Euler Loss quantifies the alignment between generated heartbeats and the dynamical model's output.

The first component of the Euler Loss quantifies how closely the temporal dynamics of each generated single lead adhere to the predictions of the EDM. For a generated heartbeat $h_k$ (for lead $k$), and its corresponding EDM parameters $\boldsymbol{\eta}_k$, this is defined as:

$$\Delta_{\text{EDM}}(h_k, \boldsymbol{\eta}_k) = \sum_{\ell=1}^{L-1} \left[ \frac{h_{k,\ell+1} - h_{k,\ell}}{\Delta t} - F_z\big(x_{k,\ell}, y_{k,\ell}, h_{k,\ell}, t_\ell; \boldsymbol{\eta}_k\big) \right]^2 \tag{8}$$

In this equation, $h_k$ denotes the generated heartbeat for lead $k$, $\boldsymbol{\eta}_k$ includes the EDM parameters specific to lead $k$, and $F_z(\cdot)$ represents the ODE from EDM (Equation 3) modeling the $z$ component (heartbeat trajectory). The variables $x_{k,\ell}$ and $y_{k,\ell}$ represent the trajectories of the other two state variables, computed using the discrete Euler solution to the coupled ODEs for lead $k$. This *intra-lead* Euler Loss encourages each generated signal to remain consistent with its underlying physiological dynamics as defined by the EDM.

The intra-lead Euler Loss for the generator is defined as:

$$L_G^{\text{EUL-1}}(\phi_G) = \mathbb{E}_{\mathbf{m}\sim p(\mathbf{m}),\ \boldsymbol{\eta}\sim p(\boldsymbol{\eta}|c)} \left[ \sum_{k=1}^{12} \Delta_{\text{EDM}}(G_k(\mathbf{m}), \eta_k) \right] \tag{9}$$

Here, the expectation is taken over noise inputs $\mathbf{m}$ and EDM parameters $\boldsymbol{\eta}_k$, with $\eta_k$ sampled from a class-conditional Gaussian $p(\boldsymbol{\eta} \mid c)$ for lead $k$ and class $c$. The conditioning vector $c$ includes the diagnostic label and RR-interval duration.

These parameters are precomputed by fitting the McSharry et al. (2003) model to real beats for each class and lead. This modeling ensures that each generated heartbeat not only exhibits realistic morphology, but also conforms to the underlying physiological dynamics prescribed by the EDM for its specific lead and condition. The Euler loss is fully differentiable, enabling end-to-end backpropagation through the generator.

**Incorporating Inter-Lead Dependencies:** A critical aspect of generating truly realistic 12-lead ECGs is not only accurate modeling of individual leads but also the correct representation of their physiological inter-dependencies. By leveraging standard ECG lead configurations and relationships derived from Einthoven's triangle and Goldberger's central terminal (Keener & Sneyd, 2009), our generator is further constrained to maintain these physiological relationships. These inter-lead dependencies are crucial for realistic multi-lead ECG synthesis and clinical interpretability.

For instance, the fundamental relationships between the limb leads (I, II, III) and augmented limb leads (aVR, aVL, aVF) are:

$$I = II - III, \qquad II = I + III, \qquad III = II - I,$$
$$aVR = -\frac{1}{2}(I + II), \quad aVL = \frac{1}{2}(I - III), \quad aVF = \frac{1}{2}(II + III) \tag{10}$$

To explicitly enforce these inter-lead dependencies, we introduce a second component of the Euler Loss that constrains the temporal derivative of each generated dependent lead to match the weighted sum of the derivatives predicted by the EDM for its constituent leads. In practice, we approximate the time derivative of a dependent lead, such as Lead II, as the sum of the derivatives of its corresponding limb leads evaluated under their respective EDM parameters. For example, the inter-lead distance for Lead II is defined as:

$$\Delta_{\text{EDM}}\big(h_{II}, \eta_I, \eta_{III}\big) = \sum_{\ell=1}^{L-1} \left[ \frac{h_{II,\ell+1} - h_{II,\ell}}{\Delta t} - (F_z(\cdot; \eta_I) + F_z(\cdot; \eta_{III})) \right]^2 \tag{11}$$

This formulation applies for all leads where such a linear combination exists, with coefficients ($\beta, \gamma$, etc.) determined by Equation 10. The parameters $\eta_I, \eta_{III}$ denote the EDM parameters associated with the constituent leads. This term encourages the generated signals to strictly adhere to physiological inter-lead relationships dictated by the underlying ODE model.

The inter-lead Euler Loss for the generator is defined as:

$$L_G^{\text{EUL-2}}(\phi_G) = \mathbb{E}_{\mathbf{m} \sim p(\mathbf{m}),\ \eta \sim p(\eta|c)} \left[ \sum_{k \in \mathcal{L}_{\text{dependent}}} \Delta_{\text{EDM}}^{\text{Inter}}(G_k(\mathbf{m}), \eta) \right] \tag{12}$$

where $\mathcal{L}_{\text{dependent}}$ denotes the set of leads that are linear combinations of others).

By leveraging inter-lead dependencies, we enforce physiological constraints within the generative model so that synthetic ECGs maintain correct relationships among leads. The generator is trained with both standard and ODE-based losses, minimizing deviations from mathematically derived expectations.

The final Euler Loss is a composite measure designed to guide the generator toward producing heartbeats that not only adhere to the temporal dynamics of individual leads (via $L_G^{\text{EUL-1}}$) but also strictly respect the inter-lead dependencies (via $L_G^{\text{EUL-2}}$):

$$L_G^{\text{EUL}}(\phi_G) = \delta \cdot L_G^{\text{EUL-1}}(\phi_G) + (1 - \delta) \cdot L_G^{\text{EUL-2}}(\phi_G) \tag{13}$$

where $\delta \in [0, 1]$ is a hyperparameter that balances the contributions of the individual and inter-lead Euler Loss components (set to 0.6, see Exp. 6.2).

The overall generator loss combines the WGAN term and the Euler Loss:

$$L_G(\phi_G) = -\mathbb{E}_{z \sim p_z}[D_w(G(z))] + \lambda_{\text{EUL}} L_G^{\text{EUL}}(\phi_G). \tag{14}$$

Figure 1 illustrates the generator–discriminator loop and the two Euler-loss paths.

### 4.3 MultiODE-GAN Discriminator

The discriminator in the MultiODE-GAN framework retains the architecture and optimization strategy typical of WaveGAN. Its primary role is to differentiate between authentic 12-lead ECG heartbeats derived from real patients and those synthetically generated by the model. To accomplish this, the discriminator is designed to maximize the Wasserstein distance, thereby enhancing its ability to identify discrepancies between the distributions of real and synthetic samples.

This formulation aligns with the standard WGAN discriminator loss as defined in Equation 7, optimizing the discriminator's ability to distinguish between data distributions effectively.

## 5 Experimental Framework

### 5.1 ECG Dataset

Our primary experiments use the Georgia 12-Lead ECG Challenge (G12EC) dataset (Alday et al., 2020), which comprises 10,344 12-lead ECG recordings from 7,871 patients in the southeastern United States. Each recording is 10 seconds long, sampled at $500\,\mathrm{Hz}$ (5,000 time steps per lead), and may contain one or more of 27 non-mutually exclusive diagnoses, enabling a comprehensive assessment of cardiac conditions.

In addition, we conduct ablation studies using the PTB-XL dataset (Wagner et al., 2020), a widely recognized benchmark for ECG classification research. Both datasets are sampled at $500\,\mathrm{Hz}$. Following established best practices in medical machine learning (Xu & Goodacre, 2018), we perform patient-level data splits, assigning 80% of patients to the combined training and validation sets and reserving the remaining 20% exclusively for testing.

### 5.2 ECG Classifier

To quantitatively evaluate the clinical utility and generative quality of the synthetic 12-lead ECG data produced by MultiODE-GAN, we assess its impact on downstream classification performance. Following standard methodologies in generative model evaluation for medical data (Golany et al., 2020; Alcaraz & Strodthoff, 2023), we compare a baseline classifier trained exclusively on real data against a classifier trained on a combination of real and synthetic samples. Any significant drop in performance on a held-out, unseen real test set would suggest a potential distributional shift or lack of fidelity in the synthetic data, thereby serving as an indirect, yet critical, measure of generative quality.

For benchmarking MultiODE-GAN, we employ two state-of-the-art architectures widely adopted for 12-lead ECG classification:

**ResNet Based:** This architecture is based on the robust ResNet model (Attia et al., 2019; Ribeiro et al., 2020), known for its effectiveness in medical signal processing. The specific implementation follows the design proposed in (Ribeiro et al., 2020). It commences with an initial convolutional layer, followed by batch normalization and ReLU activation. The core of the network consists of five residual blocks. Each block comprises three convolutional layers, interleaved with batch normalization, ReLU activation, and dropout layers, and incorporates identity skip connections to mitigate vanishing gradients. Temporal resolution is progressively downsampled via strided convolutions, applied from the second block onward, while the number of filters systematically increases across subsequent blocks. A global average pooling layer aggregates temporal features before feeding into a sigmoid-activated dense layer for multi-label binary classification.

**Attention-Enhanced ResNet:** This model incorporates ResNet backbone with a multi-head attention mechanism (Nejedly et al., 2021). The integration of multi-head attention after the convolutional layers enables the model to effectively capture long-range temporal dependencies, which is particularly beneficial for interpreting multi-lead ECG signals. This architecture was among the top-performing models in the PhysioNet Challenge, underscoring its strong discriminative power.

We train all classifiers using the Adam optimizer with learning rate $1 \times 10^{-4}$, batch size 64, for up to 100 epochs with early stopping based on validation AUC. For each abnormality, a fixed target sensitivity is selected on the validation set and the corresponding threshold is then applied unchanged across all generative-model augmentations when evaluating specificity on the test set.

### 5.3 Implementation Details

We segment each ECG recording into individual heartbeat cycles using NeuroKit2 (Makowski et al., 2021). R-peaks are detected on Lead II, and all 12 leads are temporally aligned to these peaks to ensure cross-lead coherence. Each ECG $x \in \mathbb{R}^{12 \times 5000}$ is thus converted into a set of heartbeat segments $\hat{x} \in \mathbb{R}^{12 \times L}$, where $L$ denotes the fixed beat length.

The segment length $L$ is fixed across all beats to ensure compatibility with the model architecture. In all experiments, we use a fixed length of $L = 300$ samples per heartbeat. Beats longer than $L$ are center-cropped, while shorter beats are zero-padded during preprocessing.

To avoid ambiguous beat-level labels, we restrict the main experiments to recordings in which the diagnosed abnormality is present throughout the entire 10-second signal. Consequently, all segmented heartbeats inherit the label of the original recording. Importantly, segmentation is performed *after* the patient-level data split, ensuring that no patient's recordings appear in more than one of the training, validation, or test sets, and thereby preventing data leakage.

## 6 Results and Comparative Analysis

Table 1: Classifier Performance Comparison Across Different Generative Models. Boldface indicates specificity significantly higher than the Baseline CLS at $p < 0.05$ using a 5-fold cross-validated paired t-test.

| Abnormality | Baseline CLS* | | DCGAN | WaveGAN | ME-GAN | SimGAN | SSSD-ECG | MultiODE-GAN |
|---|---|---|---|---|---|---|---|---|
| | Sensitivity | Specificity | Specificity | Specificity | Specificity | Specificity | Specificity | Specificity |
| IAVB | 0.94 | $0.82 \pm 0.018$ | $0.82 \pm 0.012$ | $0.83 \pm 0.011$ | $0.83 \pm 0.013$ | $0.83 \pm 0.013$ | $0.84 \pm 0.014$ | $\mathbf{0.85 \pm 0.012}$ |
| IRBBB | 0.82 | $0.85 \pm 0.011$ | $0.86 \pm 0.013$ | $0.87 \pm 0.011$ | $0.86 \pm 0.012$ | $0.87 \pm 0.014$ | $0.87 \pm 0.013$ | $\mathbf{0.89 \pm 0.013}$ |
| RBBB | 0.94 | $0.89 \pm 0.013$ | $0.89 \pm 0.012$ | $0.89 \pm 0.011$ | $0.90 \pm 0.010$ | $0.90 \pm 0.012$ | $0.90 \pm 0.012$ | $\mathbf{0.92 \pm 0.011}$ |
| LBBB | 0.97 | $0.96 \pm 0.002$ | $0.95 \pm 0.003$ | $0.96 \pm 0.002$ | $0.95 \pm 0.002$ | $0.96 \pm 0.002$ | $0.96 \pm 0.002$ | $0.96 \pm 0.003$ |
| NSIVCB | 0.78 | $0.72 \pm 0.015$ | $0.70 \pm 0.013$ | $0.73 \pm 0.011$ | $0.74 \pm 0.010$ | $0.74 \pm 0.010$ | $0.75 \pm 0.012$ | $\mathbf{0.76 \pm 0.014}$ |
| LAnFB | 0.89 | $0.76 \pm 0.006$ | $0.76 \pm 0.005$ | $0.77 \pm 0.006$ | $0.77 \pm 0.006$ | $0.77 \pm 0.007$ | $0.77 \pm 0.006$ | $\mathbf{0.80 \pm 0.008}$ |
| LAD | 0.89 | $0.88 \pm 0.015$ | $0.87 \pm 0.014$ | $0.89 \pm 0.011$ | $0.90 \pm 0.013$ | $0.90 \pm 0.014$ | $0.90 \pm 0.010$ | $\mathbf{0.91 \pm 0.012}$ |
| QAb | 0.81 | $0.70 \pm 0.009$ | $0.70 \pm 0.008$ | $0.70 \pm 0.009$ | $0.70 \pm 0.008$ | $0.72 \pm 0.010$ | $0.72 \pm 0.009$ | $0.72 \pm 0.008$ |
| AFL | 0.93 | $0.83 \pm 0.011$ | $0.82 \pm 0.013$ | $0.84 \pm 0.012$ | $0.85 \pm 0.014$ | $0.84 \pm 0.014$ | $0.85 \pm 0.012$ | $\mathbf{0.87 \pm 0.014}$ |

\* Baseline classifier follows Ribeiro et al. (2020)

We empirically evaluate MultiODE-GAN on 12-lead ECG classification tasks, comparing (1) a baseline classifier trained exclusively on real ECG data and (2) classifiers trained on the same real data augmented with synthetic ECG samples from various generative models, including MultiODE-GAN.

### 6.1 Classification Performance and Generative Model Comparison:

Table 1 summarizes the sensitivity and specificity metrics for a ResNet-based ECG classifier (Ribeiro et al., 2020), evaluated across multiple cardiac conditions. In all experiments, we set the classifier sensitivity to a predetermined threshold and compare the corresponding specificity. All performance metrics are assessed on a held-out test set composed entirely of unseen real samples, ensuring an unbiased evaluation of generalization. Sensitivity and specificity are standard, clinically relevant metrics widely established for such diagnostic classification tasks (Bressman et al., 2020; Wang et al., 2018; Golany et al., 2021).

To systematically evaluate the impact of different generative models on downstream classifier performance, we compare classifiers trained on datasets augmented by synthetic data from the following methods:

- **Baseline CLS (Real Data Only)** (Ribeiro et al., 2020): ResNet-based classifier trained only on real data.

- **DCGAN** (Radford et al., 2016): Convolutional GAN adapted to 12-lead ECG waveforms.

- **WaveGAN** (Donahue et al., 2018): Originally developed for audio waveform generation, WaveGAN captures temporal dependencies using convolutional structure, making it suitable for time-series data like ECGs.

- **ME-GAN** (Chen et al., 2022): A disease-aware generative model that synthesizes multi-lead ECGs via 3D-aware panoptic representations to enforce cross-lead consistency.

- **SimGAN** (Golany et al., 2020): This refers to our extended version of the original SimGAN framework. The original SimGAN was an EDM-guided GAN designed only for single-lead ECG generation and did not incorporate inter-lead dependencies. For this comparative analysis, our "extended" SimGAN applies the single-lead ODE-based loss independently to each of the 12 leads, without inter-lead relationships. This serves as a crucial baseline to demonstrate the necessity of our inter-lead constraints.

- **SSSD-ECG** (Alcaraz & Strodthoff, 2023): A recent diffusion-based model specifically designed for multi-lead ECG generation, representing the current data-driven state-of-the-art.

- **MultiODE-GAN**: Our proposed ODE-based generative adversarial network. EDM-guided GAN, incorporating both intra-lead temporal consistency and inter-lead physiological coherence.

Our results, as detailed in Table 1, consistently demonstrate that augmenting training datasets with synthetic ECGs generated by MultiODE-GAN leads to significant improvements in specificity while robustly maintaining or slightly improving sensitivity across nearly all evaluated abnormalities. These gains are particularly pronounced in lower-prevalence classes, where the limited availability of real examples often hinders the generalization capabilities of baseline models. For highly prevalent conditions, such as Left Bundle Branch Block (LBBB), where baseline models already achieve over 96% in both sensitivity and specificity, the room for further improvement is inherently limited. Sensitivity is fixed via validation-set threshold selection.

Crucially, MultiODE-GAN consistently outperforms all other generative models, including SSSD-ECG (Alcaraz & Strodthoff, 2023), which represents the current diffusion-based state-of-the-art. While diffusion models excel at capturing complex data distributions, their purely data-driven nature often lacks explicit mechanisms to enforce underlying physical or physiological laws. In contrast, our results show that MultiODE-GAN, by directly integrating biophysical ODE constraints and enforcing inter-lead physiological relationships, achieves performance that is not only comparable to, but often superior to, this strong diffusion baseline.

In all experiments, we augmented the real training data with synthetic samples at a fixed ratio of $N$ real samples to $2N$ synthetic samples. This ratio was kept constant across all generative models.

**Statistical Significance** We performed 5-fold cross-validation and computed specificity per fold. For each abnormality, we conducted a paired t-test comparing the performance of the baseline classifier against the classifiers augmented with synthetic data. All p-values were $< 0.05$, indicating that the improvements of MultiODE-GAN over the Baseline CLS are statistically significant.

## 6.2 Ablation Studies

We next perform ablation studies to quantify the contribution of key components of MultiODE-GAN to classifier performance. All experiments were conducted while maintaining a constant target sensitivity across different conditions, allowing us to precisely quantify the changes in specificity achieved by each modification. Additional experiments including quantitative analysis of inter-lead dependencies, the effect of the number of generated samples, and real-world evaluation are provided in Appendix A and Appendix B.

**Impact of Base Classifier Architectures** This experiment evaluates how different base classifier architectures affect performance when trained with synthetic data generated by MultiODE-GAN. We specifically examine the performance impact when augmenting the training data for an alternative classifier, as detailed by (Nejedly et al., 2021) from the PhysioNet/CinC Challenge. This classifier incorporates a ResNet backbone enhanced with a multi-head attention mechanism, known for its ability to capture long-range temporal dependencies. Table 2 presents the results, showing a significant improvement in the classifier's specificity when synthetic data generated by MultiODE-GAN is included in the training process.

This improvement underscores the value of our generated synthetic data in enhancing the robustness and performance of 12-lead ECG classifiers. By providing diverse and high-quality training samples, our approach helps classifiers generalize better, especially in scenarios where real-world data is scarce or imbalanced.

Table 2: Performance comparison (Sensitivity/Specificity) of an attention-enhanced ResNet classifier (Nejedly et al., 2021) trained with and without augmentation by MultiODE-GAN synthetic data on the G12EC test set. Bold indicates statistically significant improvement.

| Abnormality | Alt-CLS* | | +MultiODE-GAN | |
|---|---|---|---|---|
| | Sensitivity | Specificity | Sensitivity | Specificity |
| IAVB | 0.93 | 0.85 | 0.93 | **0.89** |
| IRBBB | 0.81 | 0.84 | 0.81 | **0.88** |
| RBBB | 0.93 | 0.90 | 0.93 | **0.92** |
| LBBB | 0.96 | 0.96 | 0.96 | **0.97** |
| NSIVCB | 0.80 | 0.73 | 0.80 | **0.77** |
| LAnFB | 0.90 | 0.76 | 0.90 | **0.80** |
| LAD | 0.91 | 0.87 | 0.91 | **0.90** |
| QAb | 0.83 | 0.72 | 0.83 | **0.75** |
| AFL | 0.92 | 0.82 | 0.92 | **0.85** |

\* Alternative Classifier architecture from Nejedly et al. (2021).

**Impact of Lead-Dependent Loss** This experiment investigates the influence of the novel lead-dependent Euler loss function on both the quality of synthetic ECG data and the performance of downstream classifiers. As defined in Section 4, the Euler loss consists of complementary intra- and inter-lead components, $L_G^{\text{EUL-1}}$ and $L_G^{\text{EUL-2}}$. The objective of $L_G^{\text{EUL-1}}$ is to ensure that the temporal dynamics of each generated lead closely follow its corresponding real lead according to the physiological dynamical model, thereby maintaining alignment with the model's equations. Conversely, $L_G^{\text{EUL-2}}$ captures the crucial inter-lead dependencies, ensuring that each generated lead accurately reflects its physiological relationships with other leads.

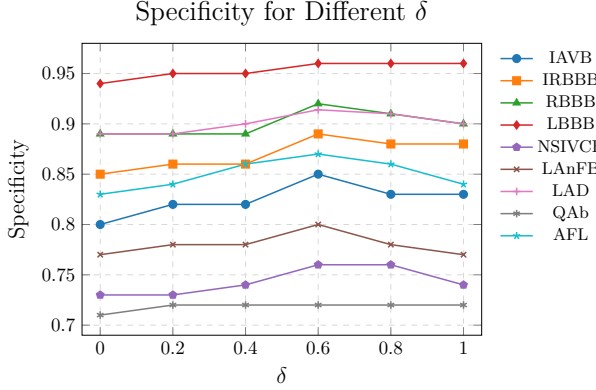

Figure 2: Classifier specificity as a function of the $\delta$ weighting parameter in the composite Euler Loss (Eq. 13). The optimal value at $\delta = 0.6$ demonstrates the necessity of balancing individual lead temporal accuracy with inter-lead physiological consistency.

We varied the weighting parameter $\delta \in [0, 1]$, where $\delta = 1$ uses only $L_G^{EUL-1}$ and $\delta = 0$ uses only $L_G^{EUL-2}$. As shown in Figure 2, the optimal performance occurs at $\delta = 0.6$, highlighting the critical importance of balancing individual lead accuracy with comprehensive inter-lead consistency. Notably, performance significantly degrades when either component is omitted (i.e., $\delta = 0$ or $\delta = 1$), underscoring the complementary and essential role of both terms in producing high-fidelity, physiologically realistic 12-lead ECGs.

**Evaluation on the PTB-XL Dataset** To assess the generalizability of MultiODE-GAN, we extended our analysis to the PTB-XL dataset (Wagner et al., 2020), a well-known large-scale benchmark comprising 21,799 12-lead ECGs from 18,869 patients. Each 10-second recording includes rich metadata and diagnostic labels.

Following the experimental setup in Section 5, we compared a baseline classifier trained exclusively on real PTB-XL data with one augmented using synthetic ECGs generated by MultiODE-GAN. As shown in Table 3, incorporating synthetic data led to consistent gains in classification specificity, particularly for rare or complex cardiac conditions.

Table 3: PTB-XL - Classifier performance across multiple evaluation metrics. Results are shown for the baseline classifier trained on real data only and with augmentation by MultiODE-GAN synthetic data. Boldface indicates statistically significant improvement over the Baseline CLS.

| Abnormality | Baseline CLS* | | | | + MultiODE-GAN | | | |
|---|---|---|---|---|---|---|---|---|
| | Sens. | Spec. | AUROC | AUPRC | Sens. | Spec. | AUROC | AUPRC |
| IAVB | 0.92 | 0.83 | 0.897 | 0.313 | 0.92 | **0.85** | **0.918** | **0.356** |
| IRBBB | 0.91 | 0.86 | 0.938 | 0.652 | 0.91 | **0.90** | **0.966** | **0.694** |
| LBBB | 0.97 | 0.97 | 0.994 | 0.906 | 0.97 | 0.97 | 0.993 | 0.907 |
| NSIVCB | 0.81 | 0.71 | 0.750 | 0.181 | 0.81 | **0.76** | **0.786** | **0.229** |
| LAnFB | 0.96 | 0.87 | 0.974 | 0.800 | 0.96 | **0.89** | **0.982** | **0.836** |
| LAD | 0.93 | 0.87 | 0.936 | 0.827 | 0.93 | **0.88** | **0.944** | **0.851** |
| QAb | 0.78 | 0.70 | 0.814 | 0.204 | 0.78 | **0.73** | **0.839** | **0.241** |
| AFL | 0.89 | 0.78 | 0.907 | 0.412 | 0.89 | **0.83** | **0.933** | **0.459** |

* Baseline classifier follows Ribeiro et al. (2020).

## 6.3 Quantitative Analysis of Inter-Lead Dependencies

To quantitatively validate the ability of MultiODE-GAN to capture and enforce physiological inter-lead dependencies, we analyze the consistency between generated leads and their expected linear combinations. Specifically, we compute the Mean Squared Error (MSE) between a generated dependent lead (e.g., Lead I) and the sum of its physiologically constituent leads (e.g., Lead II − Lead III), as defined by Einthoven's law. This metric directly assesses how well the generated signals adhere to fundamental anatomical relationships.

Table 4 presents the average MSE values for key dependent leads generated by MultiODE-GAN compared to baseline generative models. Lower MSE values indicate a stronger adherence to physiological inter-lead relationships. MultiODE-GAN consistently achieves significantly lower inter-lead MSE compared to other models, providing quantitative evidence that its $L_G^{\text{EUL-2}}$ component effectively guides the generator to produce physiologically coherent multi-lead ECGs.

Table 4: Mean Squared Error (MSE) for inter-lead dependencies for heartbeats generated by different models. Lower MSE indicates better physiological consistency.

| Model | Lead I (II−III) | Lead II (I+III) | Lead III (II−I) | aVR −0.5×(I+II) | aVL 0.5×(I−III) | aVF 0.5×(II+III) |
|---|---|---|---|---|---|---|
| DCGAN | 0.085 | 0.091 | 0.088 | 0.079 | 0.090 | 0.088 |
| WaveGAN | 0.078 | 0.083 | 0.080 | 0.071 | 0.083 | 0.080 |
| ME-GAN | 0.062 | 0.069 | 0.066 | 0.058 | 0.068 | 0.065 |
| SSSD-ECG | 0.055 | 0.060 | 0.058 | 0.050 | 0.060 | 0.057 |
| **MultiODE-GAN** | **0.041** | **0.044** | **0.042** | **0.038** | **0.051** | **0.046** |

### 6.4 Additional Evaluation

**Distributional Alignment.** In addition to classification accuracy, we quantitatively assess how closely MultiODE-GAN approximates the real data distribution. We compute the Kullback–Leibler (KL) divergence and Maximum Mean Discrepancy (MMD) between real and synthetic ECGs. To provide a clinically relevant comparison, these metrics are calculated in a learned feature space derived from a pre-trained ResNet-based encoder. Lower values for both KL divergence and MMD indicate better alignment between the real and generated data distributions. MultiODE-GAN achieves consistently low KL ($< 0.5$) and MMD ($< 0.01$) scores across all abnormalities, underscoring its ability to generate realistic, high-fidelity data that closely mirror the statistical properties of real ECGs.

**Qualitative Results** Figure 3 provides a visual illustration of the high fidelity of ECG heartbeats generated by MultiODE-GAN. The figure displays synthetic samples for Leads II and V1, meticulously overlaid with corresponding real ECG signals.

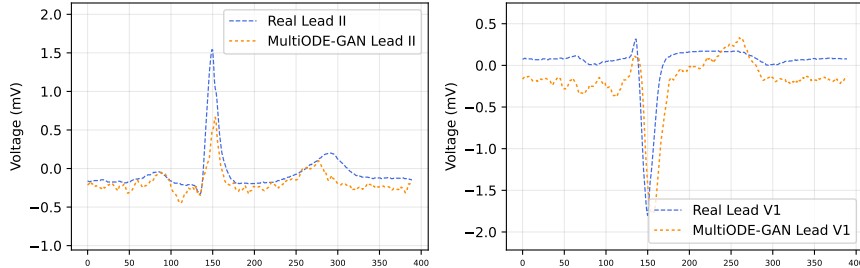

Figure 3: Representative synthetic ECG heartbeats generated by MultiODE-GAN (orange) overlaid with real ECG heartbeats (blue) for Leads II and V1.

**Evaluation Metrics.** In addition to sensitivity and specificity at fixed operating points, we report threshold-independent metrics to provide a more comprehensive assessment of downstream classifier performance. Specifically, we evaluate the area under the receiver operating characteristic curve (AUROC) and the area under the precision–recall curve (AUPRC), which are standard metrics for imbalanced medical classification tasks.

Table 5: Classifier performance comparison across different generative models on the G12EC dataset, evaluated using AUROC and AUPRC.

| Abnormality | Baseline CLS* AUROC / AUPRC | DCGAN AUROC / AUPRC | WaveGAN AUROC / AUPRC | ME-GAN AUROC / AUPRC | SimGAN AUROC / AUPRC | SSSD-ECG AUROC / AUPRC | MultiODE-GAN AUROC / AUPRC |
|---|---|---|---|---|---|---|---|
| IAVB | 0.921 / 0.475 | 0.923 / 0.487 | 0.928 / 0.501 | 0.929 / 0.508 | 0.930 / 0.509 | 0.934 / 0.521 | **0.941 / 0.558** |
| IRBBB | 0.902 / 0.386 | 0.908 / 0.401 | 0.912 / 0.414 | 0.910 / 0.409 | 0.914 / 0.418 | 0.916 / 0.423 | **0.928 / 0.481** |
| RBBB | 0.973 / 0.749 | 0.973 / 0.751 | 0.974 / 0.758 | 0.975 / 0.764 | 0.975 / 0.766 | 0.974 / 0.772 | **0.982 / 0.801** |
| LBBB | 0.989 / 0.851 | 0.988 / 0.848 | 0.989 / 0.853 | 0.988 / 0.850 | 0.989 / 0.855 | 0.989 / 0.856 | **0.990** / 0.859 |
| NSIVCB | 0.834 / 0.140 | 0.836 / 0.149 | 0.842 / 0.158 | 0.846 / 0.164 | 0.844 / 0.162 | 0.848 / 0.169 | **0.856 / 0.195** |
| LAnFB | 0.921 / 0.266 | 0.922 / 0.273 | 0.925 / 0.281 | 0.926 / 0.283 | 0.926 / 0.285 | 0.927 / 0.287 | **0.936 / 0.342** |
| LAD | 0.944 / 0.638 | 0.943 / 0.641 | 0.946 / 0.653 | 0.948 / 0.661 | 0.948 / 0.663 | 0.949 / 0.668 | **0.956 / 0.691** |
| QAb | 0.725 / 0.181 | 0.726 / 0.184 | 0.728 / 0.188 | 0.729 / 0.190 | 0.733 / 0.195 | 0.732 / 0.197 | **0.734 / 0.205** |
| AFL | 0.856 / 0.303 | 0.855 / 0.306 | 0.858 / 0.315 | 0.861 / 0.321 | 0.860 / 0.319 | 0.863 / 0.325 | **0.881 / 0.377** |

* Baseline classifier follows Ribeiro et al. (2020).

## 7 Limitations

While MultiODE-GAN demonstrates significant advancements in physiologically-informed ECG synthesis, certain limitations and avenues for future research exist. Integrating ODEs within the GAN training loop is computationally intensive, requiring further optimization for large-scale or real-time applications. Moreover, pre-computation of EDM parameters necessitates some real labeled data, posing a challenge for generating entirely novel cardiac conditions. For abnormalities where baseline classifiers already achieve near-perfect performance (e.g., LBBB), the incremental gains from synthetic data augmentation are naturally constrained.

# 8 Conclusions

In this paper, we introduced MultiODE-GAN, a novel generative adversarial framework for synthesizing highly realistic 12-lead ECG signals by incorporating domain-specific knowledge from the ECG Dynamical Model. Our approach leverages a novel Euler Loss to enforce both intra-lead temporal dynamics and critical inter-lead physiological dependencies, yielding clinically plausible and structurally coherent ECG waveforms.

Our empirical results show that MultiODE-GAN consistently outperforms conventional generative methods, including state-of-the-art diffusion models. This superior generative quality translates into significant gains in downstream classification performance, particularly in specificity for lower-prevalence cardiac abnormalities especially in lower-prevalence classes. Furthermore, we validated MultiODE-GAN's clinical utility in a real-world setting, showing its potential for early detection of left ventricular systolic dysfunction (LVSD) by identifying patients potentially in need of further echocardiographic evaluation.

MultiODE-GAN holds significant promise for advancing cardiac diagnostics and research. Future work will explore diffusion-guided ODEs to combine the strengths of both generative paradigms for even higher fidelity and diversity. We also plan to extend its application to a broader spectrum of cardiac conditions and explore its utility in personalized analysis to further support clinical decision-making and improve patient outcomes.

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

# A Additional Ablation Studies

## A.1 Generated Samples Number

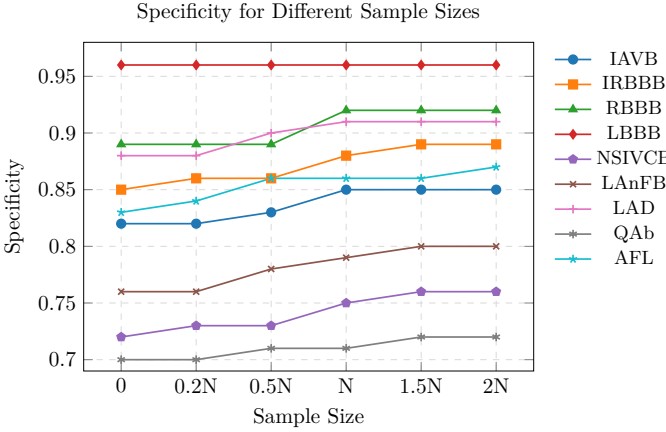

Figure 4: Classifier specificity across different synthetic sample sizes (0.2*N*, 0.5*N*, *N*, 1.5*N*, 2*N*) for each abnormality class. Results show synthetic data improves specificity, with peak gains at optimal sample sizes.

This experiment explores how the quantity of synthetic data samples generated by the MultiODE-GAN model affects classifier performance. We analyze the effect of varying synthetic sample sizes on classifier specificity and generalization capabilities. Specifically, for each class with N existing samples, we generate synthetic datasets of sizes 0.2N, 0.5N, N, 1.5N, and 2N. Figure 4 illustrates that adding synthetic samples generally improves specificity, with certain sample sizes yielding the maximum improvement, while sensitivity remains constant.

Figure 4 illustrates that adding synthetic samples generally improves specificity, with certain sample sizes yielding the maximum improvement, while sensitivity remains constant. This analysis provides practical guidance on the optimal amount of synthetic data to augment real datasets for maximal performance gains.

## B Real-World Evaluation: LVSD Prediction in Clinical Practice

To evaluate the practical utility of MultiODE-GAN in a real-world clinical context, we conducted an experimental study in collaboration with the Interventional Cardiology Unit, focusing on the early detection of left ventricular systolic dysfunction (LVSD) directly from standard 12-lead ECG data.

Early detection of left ventricular systolic dysfunction (LVSD), a precursor to heart failure, is crucial for timely intervention. Current clinical practice relies on echocardiography to estimate ejection fraction (EF), with LVSD defined as EF < 50% (McDonagh et al., 2021).Ejection fraction is a key measurement used to assess the heart's pumping efficiency, representing the percentage of blood ejected from the left ventricle. However, echocardiography is costly, requires specialized expertise, and is not always accessible in primary care settings (Rutten et al., 2003; Hobbs et al., 2000). In contrast, 12-lead ECG is ubiquitous, non-invasive, and cost-effective making it an ideal alternative for initial LVSD screening.

### B.1 Clinical Dataset

We used a dataset of 13,820 paired ECG and echocardiogram records, with one ECG-echocardiography pair per patient. The data was partitioned into training (10,402), validation (2,568), and test (854) sets. The test set was balanced to include an equal proportion of normal and abnormal LV function cases, with abnormal LV function defined as EF < 50%. . Each ECG was resampled to 500 Hz and formatted as a $12 \times 5000$ matrix (12 leads, 10 seconds). Short signals were zero-padded, longer signals truncated.

**Ethics Statement:** All data collection were approved by the institutional IRB at the participating medical center, with informed consent obtained and all data de-identified prior to analysis.

### B.2 Model Architecture

Our ResNet-based classifier, adapted from (Attia et al., 2019; Ribeiro et al., 2020), receive a $12 \times 5000$ ECG matrix as input. The architecture begins with a convolutional layers, followed by batch normalization and ReLU activation. It then passes through five residual blocks, each comprising three convolutional layers with batch normalization, ReLU, dropout, and skip connections to promote stable gradient flow.

After the final residual block, a global average pooling layer aggregates the learned features, which are passed to a dense layer with sigmoid activation for binary classification (EF < 50% vs. EF $\geq$ 50%). The model was trained using the Adam optimizer with an initial learning rate of 0.0001, minimizing binary cross-entropy loss. We conducted iterative hyperparameter tuning across optimizer types (Adam, RMSProp, SGD), learning rates $(0.01, 0.001, 0.0001)$, dropout rates, and residual block depths (3–7), selecting the best configuration based on validation accuracy. Training was performed for up to 100 epochs with early stopping based on validation performance.

### B.3 Clinical Evaluation and Impact of Synthetic Data

Six cardiologists independently annotated the 854-patient test set, including three senior cardiologists (12–20+ years experience). Table 6 summarizes performance comparisons.

To assess the contribution of synthetic ECGs generated by MultiODE-GAN, we retrained the model using a combined dataset of real and synthetic signals, Synthetic ECGs were used exclusively during training for data augmentation. This augmentation improved specificity from 78% to 81% while maintaining sensitivity, particularly benefiting class balance and minority case coverage.

The ResNet classifier outperformed average physician accuracy and matched senior cardiologists in sensitivity (78%). Augmenting with MultiODE-GAN synthetic ECGs further improved specificity to 81%, surpassing all human benchmarks.

Table 6: LVSD detection (EF < 50%): Comparison of physician and model performance with/without MultiODE-GAN data.

|  | Sensitivity | Specificity |
|---|---|---|
| Average Physician | 0.71 | 0.68 |
| Senior Physician (avg.) | 0.77 | 0.64 |
| ResNet Classifier | **0.78** | 0.78 |
| ResNet + MultiODE-GAN | **0.78** | **0.81** |

