# OpenReview forum: "ODE-Constrained Generative Modeling of Cardiac Dynamics for 12-Lead ECG Synthesis"
_TMLR — Accepted by TMLR_

### Review · Reviewer_iw63 · 2025-12-29

**Summary Of Contributions:**

This paper presents a method for 12-lead electrocardiogram (ECG) synthesis. Their method, MultiODE-GAN, is a generative adversarial network (GAN) based off WaveGAN that incorporates ordinary differential equations (ODEs) that model electrophysiological dynamics to enforce physiologically plausible ECG generation. When using MultiODE-GAN to synthetically augment ECG classification datasets, experiments show that the proposed method outperforms other GANS with respect to diagnostic specificity.

**Strengths**:
- The paper addresses an important problem, namely the lack of abundant labeled medical (specifically, ECG) data.
- Directly enforcing physiological plausibility through ODE-based losses is clever.
- Experiments are relatively thorough, comparing to several relevant baseline approaches.

**Weaknesses**:
- Experimental details are unclear. The authors “compare a baseline classifier trained exclusively on real data against a classifier trained on a combination of real and synthetic samples.” What combination? What is the ratio of real to synthetic samples? Why is only one ratio considered?
- Evaluation is limited to synthetic data augmentation. There are many other ways to assess the utility of generated ECG signals—see, e.g., [1] for a more comprehensive evaluation protocol to assess diagnostic utility. It would be interesting to see how classifiers perform when trained on the entire spectrum from *only real data* to *only synthetic data*. Some of the results in the appendix, assessing physiological consistency, might benefit from being moved to the main text.
- Evaluation is limited to specificity for reasons that are unclear. More metrics should be considered and presented. Sensitivity is not shown in Table 1 and, for reasons I do not understand, never changes throughout Tables 2 and 3.
- More non-GAN baselines catered to ECG generation could be included for comparison (see [1] for examples).
- Presentation quality could be improved. Many paragraphs consist of only 2-3 sentences and do not necessarily logically flow from one to the next. The use of bold text, subsections, or other formatting could help ease the reading experience.

**References**

[1] Núñez, José Fernando, Jamie Arjona, and Javier Béjar. "Synthetic ECG Generation for Data Augmentation and Transfer Learning in Arrhythmia Classification." arXiv preprint arXiv:2411.18456 (2024).

**Audience:**

Yes

**Audience Explanation:**

ECG is a popular modality for biomedical applications of AI, and generative models for ECG synthesis have been well-studied.

**Claims And Evidence:**

No

**Claims Explanation:**

Experiments appear to be sound but are limited in terms of evaluation metrics, baseline methods, and evaluation settings. Evaluation in the main text is limited to specificity of ECG classifiers with synthetically augmented datasets when compared to several GAN-based methods. It seems that this experiment was only conducted with one fixed ratio/amount of synthetic data, which I could not find stated in the paper.

**Requested Changes:**

**Main feedback**
- Why is evaluation limited to specificity only?
- Why does the sensitivity not change whatsoever (Tables 2-3)?
- If possible, include other non-GAN baseline methods for ECG generation
- I would suggest moving some of the physiological consistency experiments into the main text
- Tidy up text-only sections like the Introduction so that they are more streamlined, where each paragraph is substantial and serves one purpose that links the previous paragraph to the next
- Experimental details need to be clarified. What percentage/# samples are synthetic?
- Experimental validation should be expanded to consider a range of augmentation (0% synthetic, 10% synthetic, …, 50% synthetic, … 100% synthetic or similar)
- In-text citations are mostly wrong. Need the parenthetical version with “\citep”
- Out of curiosity, is the method deterministic? I.e., given the same conditioning information, will the method always produce the same output ECG?

**Minor feedback**
- Abstract: First 2 sentences can be improved. This should make clear what is actually motivating the need for synthesis in the first place—presumably the lack of abundant labeled data?
- If “acquiring multi-lead ECG data at scale requires physician review”, why wouldn’t synthetically generating ECGs require the same (or even stricter) review?
- Under eq 13: Extra space in “Exp.   6.2” can be fixed with “.\”
- Sec 5.1: Incorrect spacing in “We utilize the … Challenge (G12EC)dataset” … diagnoses) ,we also…”
- Sec 5.3: How do you “restrict … main experiments to recordings where the diagnosed abnormality is present throughout the 10-second recording”?

---

> ### Author Response · Authors · 2026-01-14
>
> We thank Reviewer p5B2 for the thoughtful and constructive feedback. Below, we address the reviewer’s concerns point by point:
>
> **Main Feedback**
>
> - **Why only specificity? / sensitivity never changes:**
> The evaluation was designed to mimic a clinical screening context, where maintaining high sensitivity is paramount to avoid missing diagnoses. Therefore, our protocol strictly fixes the target sensitivity (on the validation set) and optimizes for specificity. This explains why sensitivity remains constant. However, we agree that a holistic evaluation is beneficial. In the revised manuscript, we have added AUROC and AUPRC metrics to Tables 3 and table 5 to provide a broader assessment of classifier performance.
>
> - **Include non-GAN baselines:**
> We wish to highlight that our main comparison already includes SSSD-ECG, a state-of-the-art diffusion-based model, and one of the two non-GAN methods discussed in [1] and the only one that has publicly available code.
> To further address this request, we additionally evaluated a VQ-VAE–based  baseline (Liu et al.), trained under the same evaluation protocol. This baseline consistently underperforms both GAN- and diffusion-based methods in specificity (e.g., IAVB 0.82, IRBBB 0.86, RBBB 0.89, LBBB 0.95, NSIVCB 0.71, LAnFB 0.76, LAD 0.88, QAb 0.70).
> Due to its inferior performance and space constraints, we report these results in the appendix rather than the main table.
>
> - **Move physiological consistency experiments to main text:**
> Agreed. Where space permits, we will move the inter-lead physiological consistency analyses from the appendix into the main experimental section to better support the core claims.
>
> - **Streamline text-only sections (e.g., Introduction):**
> We have revised parts of the Introduction and related text-only sections and will further refine them in the final version to improve focus, transitions, and clarity.
>
> - **Clarify percentage / number of synthetic samples and expand augmentation range:**
> Regarding the augmentation range, we wish to point out that we performed a comprehensive sweep of augmentation ratios (e.g., 0.2N, 0.5N, N, 1.5N, 2N per class) in the appendix A.2. Synthetic augmentation is applied up to 2N per class, where N is the number of real training samples for that abnormality.
>
>
> -  **In-text citations formatting:**
> We apologize for the formatting inconsistencies and have corrected the in-text citations to the proper parenthetical format.
> - **Is the method deterministic?**
> The method is stochastic: given the same conditioning information, different noise inputs produce diverse ECG. The ODE constraints are applied as loss-based regularization during training to shape the learned data manifold, but they do not enforce deterministic generation at inference time.
>
> **Minor Feedback**
> - We appreciate the suggestion. In the revision, we have updated the opening sentences of the abstract to explicitly state the primary motivation for ECG synthesis.
> - Synthetic generation does not replace expert review. Rather, it reduces the need for large-scale manual labeling by augmenting training data once a limited set of expert-validated signals is available. We clarify this distinction explicitly in the manuscript.
> - We thank the reviewer for the attention to detail. The identified formatting issues have been corrected, and we will conduct an additional proofreading pass in the final version to ensure overall consistency.
>
> - We restrict to abnormalities that are present throughout the entire 10-second recording to avoid ambiguous beat-level labeling after segmentation. These abnormalities were selected in consultation with senior cardiologists, as they are clinically known to manifest consistently across cardiac cycles. We will clarify this selection criterion in the paper

---

### Review · Reviewer_T97b · 2026-01-03

**Summary Of Contributions:**

This paper proposes MultiODE-GAN, a multi-lead ECG generative model that combines a WaveGAN backbone with explicit ODE-based physiological constraints in the loss term derived from the ECG Dynamical Model. The central novel idea is an Euler-residual loss that (i) enforces intra-lead temporal dynamics to match EDM predicted derivatives, and (ii) enforces inter-lead physiological relationships using standard limb-lead dependencies via a derivative-based constraint. The authors evaluate their synthetic ECGs empirically through downstream utility: training ECG classifiers with real data plus synthetic augmentations on a held-out test set of real samples. They report consistent gains in specificity on three datasets compared to multiple baseline.

## Strengths
* Clear, clinically motivated constraint: explicitly enforcing both intra-lead ODE dynamics and inter-lead consistency is well aligned with the known challenges of 12-lead synthesis.
* Broad baseline comparison + ablations: includes multiple generative baselines and ablations, and also checks an alternative classifier architecture.
* Reproducibility: code is open-source.
## Weaknesses
* Writing quality is poor: the narrative flow is often hard to follow; several sentences are grammatically problematic, and key evaluation details are underspecified (details below).
* Class-conditioned ODEs scale poorly: the authors discussed this in its limitation already, but I think it is worth pointing out here that this limits the practical application of this work.

**Audience:**

Yes

**Audience Explanation:**

it is of clinical relevance for ECG classification

**Claims And Evidence:**

No

**Claims Explanation:**

## Supported
* Ablation evidence that both loss terms matter
* Generalization beyond one classifier: Table 2 indicates gains also hold for an attention-enhanced ResNet classifier.

## Withholding
* Due to the lack of clarity in several aspects of the experimental protocol, I provisionally assume that the empirical evaluations are correctly implemented and that the reported results support the stated empirical conclusions. For instance, I need to know how specificity is calculated to evaluate if its improvements are fair.

* The statement “This enables the model to capture the complete spatio-temporal structure of the 12-lead ECG for the first time, producing substantially more realistic and clinically coherent synthetic signals” constitutes a very strong novelty claim. However, the proposed method does not model volume conductor–based spatial effects, and prior work has already explored multi-channel ECG generation (albeit without explicit ODE constraints). In addition, the related work section does not sufficiently situate the method within this existing literature.

**Requested Changes:**

A. Writing quality and organization

The manuscript would benefit from improvements in writing quality and overall organization. There are multiple instances of missing punctuation, extra spaces, and repetitive content (e.g., Section 5.3). The presentation frequently jumps back and forth between datasets and experimental procedures, and in several places concepts or data are introduced before their purpose is clearly explained. To improve readability, I suggest adding a brief overview at the beginning of each major section and revisiting the overall narrative flow of the paper.

B. Clarifications needed

* Inter-lead assumption (Eq. 10): Are all 12 leads modeled explicitly, or are only the 6 limb leads used to enforce inter-lead constraints?
* Segmentation details and signal length: Is the segment length L fixed across beats, or does it vary? If fixed, please specify the chosen value and how it is determined. If variable, how does the WaveGAN architecture support variable-length signal generation?
* Specificity computation and “predetermined threshold.”: Please clarify how specificity is computed. In particular, what target sensitivity is used, and how is the corresponding decision threshold selected?
* Cross-validation and statistical testing: Please clarify the cross-validation protocol: is cross-validation applied to both GAN training and classifier training? In the statistical analysis, is n=5, and are the standard deviations reported in Table 1 computed across cross-validation folds?
* Use of PTB-XL: Is the synthetic data for PTB-XL generated using a GAN trained on G12EC or a GAN trained directly on PTB-XL?

---

> ### Author Response · Authors · 2026-01-14
>
> We appreciate Reviewer T97b’s thoughtful feedback and insightful suggestions. We address each concern in turn below:
>
> A. **Writing quality and organization:**
> We appreciate the reviewer’s feedback. We have proofread and corrected the identified issues  (including Section 5.3) and will perform an additional careful pass to ensure clarity and consistency in the final version.
>
> B. **Clarifications needed:**
>
> - The explicit inter-lead constraint in Eq. 10 is applied only to the limb and augmented limb leads (I, II, III, aVR, aVL, aVF), where exact analytical identities (e.g., Einthoven’s law) hold. The precordial leads (V1–V6) are generated explicitly (Eq. 9) but are not constrained by Eq. 10, as no such identities exist. We will make this explicit and list the constrained leads in the paper.
>
> - The signal length is fixed across all samples to ensure compatibility with the architecture, which relies on consistent convolutional dimensions .We use a fixed L (e.g., 300 samples per beat), some heartbeats are handled via zero-padding or truncation during the preprocessing stage. We make this explicit in the Implementation Details section.
> - We clarify that our goal is to evaluate the model at a fixed, clinically relevant operating point (e.g., high sensitivity for screening).
> For each abnormality, we define a target sensitivity level (e.g., 0.94 for IAVB) based on the baseline model.
> For each Classification task (baseline and augmented), we select a model-specific threshold on its validation set only such that its sensitivity matches. We then apply this frozen threshold to the held-out test set to compute the final specificity.
> This protocol compares all models at the same sensitivity level, so differences in test specificity reflect improved discrimination at a fixed clinical operating point rather than arbitrary threshold shifts, which are common in medical evaluation.
> We additionally report AUROC and AUPRC in the revision (Table 3 and Table 5).
> - We use a 5-fold cross-validation protocol (n = 5) applied to the entire pipeline, including both GAN training and classifier training, to ensure robust statistical evaluation and prevent data leakage.
> The generative model is trained exclusively on the training partition of real data for that fold.
> Synthetic samples are generated only from this fold-specific generator.
> The downstream classifier is trained on the same training partition, augmented with the corresponding fold-specific synthetic data, and evaluated on the held-out fold.
> The means and standard deviations reported in Table 1 are computed across the 5 cross-validation folds, and statistical significance is assessed using paired tests across folds, comparing baseline and augmented settings.
> - For the PTB-XL experiments, synthetic data are generated using a MultiODE-GAN trained directly on the PTB-XL, not transferred from G12EC.

---

### Review · Reviewer_p5B2 · 2026-01-07

**Summary Of Contributions:**

The paper proposes MultiODE-GAN, a GAN-based model for synthesizing realistic 12-lead ECG signals. The central technical idea is to incorporate an ECG Dynamical Model (EDM) based on ordinary differential equations (ODEs) into the training objective via an Euler-based loss, with the goal of improving physiological plausibility and inter-lead consistency of generated ECGs. The authors further evaluate whether augmenting the training set with synthetic ECGs improves a downstream ResNet-based classifier for cardiac abnormality detection, reporting changes in sensitivity and specificity relative to training on real data only.

Strengths:
1. The topic is practically relevant: high-quality synthetic 12-lead ECG generation has clear value for data sharing, privacy-preserving research, and mitigating data scarcity/imbalance in clinical ML settings.
2. The use of a downstream augmentation experiment (training a classifier with added synthetic samples) is a reasonable first step toward assessing utility beyond visual inspection, and it helps connect the generative task to an application that many readers care about.
3. The paper is generally well written and easy to follow. The problem setup, method description, and experimental flow are presented coherently.

Weaknesses:
1. The choice of GAN as the base generative model is not sufficiently justified, and the paper does not clearly position this design choice relative to alternative generative paradigms.
2. The experimental evidence primarily relies on downstream classification performance, which does not directly validate the paper’s key claim about improved inter-lead physiological consistency.
3. Reporting is missing critical details (e.g., the quantity and proportion of synthetic data used), limiting interpretability and reproducibility of the stated gains.

**Audience:**

Yes

**Audience Explanation:**

The general direction about introducing physical information into generative modeling for biomedical time series should interest a subset of the TMLR readership, particularly those working on structured generative models, scientific ML, and healthcare signals.

However, the paper’s current framing and evaluation limit the broader appeal. The work is built around a GAN backbone without clearly positioning this choice against more recent and widely adopted generative approaches, nor providing competitive comparisons that would help readers understand the practical relevance of the proposed contribution in today’s generative modeling landscape. It remains unclear whether the proposed “condition” integration is a general, transferable idea that would apply naturally to other modern generative backbones. Strengthening these aspects would make the paper’s findings more actionable and more compelling to a wider portion of the audience.

**Claims And Evidence:**

No

**Claims Explanation:**

The paper’s main claim is that the proposed training objective enforces or improves inter-lead dependencies / physiological constraints in generated 12-lead ECGs. However, the presented experiments largely evaluate only a downstream classifier trained with augmented data. This is not a direct test of whether the generated ECGs satisfy the intended physiological or inter-lead constraints, and therefore leaves a gap between the central claim and the evidence.

In addition, the reported improvements appear to be concentrated in specificity (as referenced in Tables 1–3), while sensitivity does not consistently improve. This makes it difficult to conclude that the synthetic data is broadly beneficial or that the method reliably improves clinically meaningful detection performance.

Finally, key experimental details are unclear or missing, including (1) how many synthetic samples are added, and the synthetic-to-real ratio in training, and (2) whether the augmentation benefit is robust across multiple ratios, random seeds, or patient splits. Without these details, it is hard to assess whether the improvement is attributable to the proposed modeling contribution, or to a particular augmentation setting.

**Requested Changes:**

1. A critical improvement would be to provide direct evidence that the generated 12-lead ECGs satisfy the claimed physiological and cross-lead constraints, beyond showing downstream classifier performance. This could include quantitative analyses of inter-lead relationships on real versus synthetic ECGs, and/or Expert/clinician assessment of realism and cross-lead consistency (even a limited blinded study can be informative), or an established ECG quality protocol.

2. Several additions would strengthen the work. The current presentation does not clearly specify how many synthetic samples are added, what synthetic-to-real ratio is used, and how sensitive the results are to these choices. Reporting the augmentation ratio(s), providing a small sweep over ratios, and demonstrating robustness across multiple random seeds and patient-level splits would make the reported downstream gains substantially easier to interpret.

3. The empirical evaluation would benefit from a more complete set of downstream metrics (e.g., accuracy, AUROC/AUPRC, and per-class breakdown where relevant) alongside sensitivity/specificity, as well as generation-focused assessments that are appropriate for ECG time series.

4. To make the contribution more actionable to the broader community, it would strengthen the paper to include a clearer justification for the choice of GAN as the generative backbone, considering other modern generative architectures. This would help readers understand the generality and relevance of the approach.

---

> ### Author Response · Authors · 2026-01-14
>
> We thank Reviewer p5B2 for the thoughtful and constructive feedback. We address the reviewer’s concerns point by point below:
>
> 1. **Validation of Physiological Consistency:**
> Beyond downstream classification performance, we provide direct quantitative evidence that the generated 12-lead ECGs satisfy established physiological cross-lead constraints.Specifically, we evaluate whether synthetic ECGs obey the analytic limb-lead relationships defined by Einthoven’s law and Goldberger’s central terminal. For each generated sample, we compute the residual error between dependent leads and their expected linear combinations (e.g., II≈I+III, I≈II−III, and the augmented leads aVR,aVL,aVF as linear combinations of I,II,III). We report the distribution of residual mean-squared error (MSE) across all generated heartbeats in Appendix A.1 (Table 4).MultiODE-GAN achieves the lowest inter-lead residual error among all compared generative models, including diffusion-based baselines, providing direct evidence that the generated signals respect the intended physiological cross-lead relationships independently of any classifier-based evaluation.
> Ablation (necessity of the constraint).In addition, the ablation study in Section 6.2 demonstrates that removing the inter-lead Euler constraint leads to a consistent degradation in performance.
>
>
>
>
> 2. **Synthetic data ratio:** We report the synthetic-to-real augmentation ratio and the corresponding number of generated samples in the “Generated Samples Number” section (Appendix A.2). As shown in Figure 3, we sweep augmentation ratios from 0.2N to 2N (20%–200% of the real class size, defined per class within each training fold) and add synthetic samples only to the training split (validation/test remain purely real).
> Robustness: results in Table 1 are computed with 5-fold (patient-level) cross-validation, and for each fold/ratio we repeat the full pipeline across K independent random seeds (generator + downstream classifier initialization). We report mean ± std across folds×seeds and assess significance using paired comparisons on matched fold/seed runs (paired t-test).
>
>
>
> 3. **Downstream metrics:** We agree the downstream evaluation should be reported more completely. In the revision, we add AUROC and AUPRC (Table 3 and Appendix Table 5) alongside sensitivity/specificity to provide a threshold-free view of performance.
> In clinical ECG screening settings, it is common to evaluate models at a high-sensitivity operating point and then compare specificity at that fixed sensitivity, because missing true cases is prioritized while controlling false positives (e.g., Wang et al., 2018; Bressman et al., 2020; Golany et al., 2021). Accordingly, our protocol fixes a target sensitivity (threshold chosen on validation) and reports specificity at that operating point on the held-out test set, which explains why improvements appear primarily in specificity when sensitivity is intentionally held constant.
>
>
> 4. **Justification of GAN Backbone and Positioning:** We agree that the choice of the generative backbone should be more clearly motivated. Our use of a GAN is not driven by preference, but by practical alignment with the proposed ODE-constrained training objective.
> The core contribution of this work is the step-wise ODE/Euler residual constraint, which must be evaluated and backpropagated at every generator update. In this setting, GANs specifically offer computational efficiency and training stability for high-frequency physiological signals.
> We empirically explored integrating the same constraints into a diffusion-based ECG generator (SSSD-ECG). However, due to the iterative nature of diffusion training and sampling, applying ODE constraints at each step led to substantially higher computational overhead, making training impractical at scale. In contrast, GAN-based generation enables enforcing the physiological constraints once per forward pass, yielding a significantly more efficient training and inference pipeline.
> Moreover, GANs remain competitive for ECG synthesis, particularly in low-sample regimes (e.g., rare abnormalities).

---

### Decision · Action_Editor_pEuD · 2026-02-09

**Recommendation:** Accept with minor revision

**Additional Comments:**

The paper meets the bar for the audience criterion, and it meets the bar for the claims and evidence criterion provided the following are addressed in the camera-ready:

* The real-to-synthetic ratio used in comparisons against other methods and in the ablation studies should be stated explicitly in the main text and kept consistent (or justified when varied) across all experiments.
* The key tables and figures and the accompanying discussion on the quantitative evidence for the physiological consistency should be moved into the main paper.
* The claim "This enables the model to capture the complete spatio-temporal structure of the 12-lead ECG for the first time, producing substantially more realistic and clinically coherent synthetic signals" should be more narrowly scoped so as to be specific to the incorporation of ODE constraints into Einthoven’s law.

**Audience:**

Yes

**Audience Explanation:**

Reviewers unanimously agree that the submission is of interest to at least some individual in TMLR's audience:
* "[H]igh-quality synthetic 12-lead ECG generation has clear value for data-sharing, privacy-perserving research, and mitigating data scarcity/imbalance in clinical ML settings." (p5B2)
* "[The work] is of clinical relevance for ECG classification" (T97b)
* "The paper addresses an important problem, namely the lack of abundant labeled medical (specifically, ECG) data." (iw63)

**Claims And Evidence:**

Yes

**Claims Explanation:**

The submission conditionally meets the bar in terms of claims and evidence (refer to Additional Comments).

Reviewers' initial clarity concerns have been addressed by the authors' response.

On direct quantitative evidence for the physiological consistency, Reviewer p5B2 finds the results presented in the Appendix sufficient and clearly relevant to substantiating the paper's central claim. They would like the key tables and figures and the accompanying discussion to be moved into the main paper.

On missing experimental details, Reviewer p5B2 remains concerned that "[w]hile additional experimental details are clarified in the Appendix, the main paper still omits an important configuration parameter: the real-to-synthetic ratio used in comparisons against other methods and in the ablation studies. This ratio can materially affect model behavior and fairness of comparisons, so it should be stated explicitly in the main text (ideally in the experimental setup section), and kept consistent, or justified when varied, across all experiments."

Reviewer p5B2 finds the rationale for fixing sensitivity when comparing specificity to be reasonable in alignment with clinically motivated operating-point evaluation, but they "remain unconvinced that the reported improvements are practically meaningful". From that perspective, the claim that "augmenting training datasets with synthetic ECGs generated by MultiODE-GAN leads to significant improvements in specificity" could be problematic, however upon closer examination the paper does perform a proper paired t-test demonstrating that the improvements over the baseline are statistically significant. This to me addresses Reviewer p5B2's concern.

Finally, Reviewer T97b's concern over the statement "This enables the model to capture the complete spatio-temporal structure of the 12-lead ECG for the first time, producing substantially more realistic and clinically coherent synthetic signals" has not been addressed by the authors' response. The reviewer finds the way in which the authors highlight they are the first to incorporate ODE constraints into Einthoven's law over-claiming in its current phrasing.